# A Systematic Review on the Impact of Vaccination for Respiratory Disease on Antibody Titer Responses, Health, and Performance in Beef and Dairy Cattle

**DOI:** 10.3390/vetsci11120599

**Published:** 2024-11-27

**Authors:** Hudson R. McAllister, Bradly I. Ramirez, Molly E. Crews, Laura M. Rey, Alexis C. Thompson, Sarah F. Capik, Matthew A. Scott

**Affiliations:** 1Veterinary Education, Research, and Outreach Program, Texas A&M University, Canyon, TX 79016, USA; 2Small Animal Clinical Sciences, Texas A&M University College of Veterinary Medicine and Biomedical Science, College Station, TX 77843, USA; 3Texas A&M Veterinary Diagnostic Lab, Canyon, TX 79016, USA; 4Tumbleweed Veterinary Services, PLLC, Amarillo, TX 79159, USA

**Keywords:** cattle, vaccination, bovine respiratory disease, antibody titers

## Abstract

Bovine respiratory disease is a serious issue for beef and dairy farms, and vaccines are commonly used to prevent it. However, understanding of how these vaccines impact cattle health and performance is not well documented. This review aimed to evaluate how well these vaccines work by looking at antibody titers and overall health in beef and dairy cattle. Researchers followed strict guidelines and reviewed studies from the USA and Canada published between 1982 and October 2022 focusing on vaccines for several BRD-related pathogens. Out of over 3000 studies initially reviewed, only 101 were included, with most focusing on beef cattle. The findings showed significant differences in reporting between studies, making it difficult to combine results. The review highlights the need for more standardized reporting to better understand how these vaccines affect cattle health and performance.

## 1. Introduction

Bovine respiratory disease (BRD) is one of the costliest diseases in cattle production, with the cost of a single respiratory disease treatment having almost doubled between 1999 and 2011, from $12.59 to $23.60, respectively [1]. Bovine respiratory disease is a polymicrobial disease complex that involves several viral and bacterial infectious agents. Common viral pathogens related to BRD are bovine viral diarrhea virus (BVDV), bovine herpesvirus-1 (BHV-1), parainfluenza 3 (PI3), and bovine respiratory syncytial virus (BRSV). Additionally, common bacterial pathogens related to BRD include *Mannheimia haemolytica* (Mh), *Pasteurella multocida* (Pm), and *Histophilus somni* (Hs) [2]. Various non-infectious components in cattle management and production are known to influence BRD development, such as novel facility enrollment, transportation, commingling, and significant environmental changes [3]. The multifactorial nature of BRD makes diagnosis and treatment of BRD difficult due to the large number of factors that influence management decisions [4].

BRD prevention is an ideal goal in cattle production systems and vaccination is a common tactic used towards that end. Approximately 95% of US feedlots administer respiratory vaccines for BVDV, BHV-1, PI3, or BRSV.1 However, only approximately 50% of cow-calf operations in the US vaccinate for respiratory-related pathogens across all cattle types [5]. While vaccination is a commonly used prevention method, the efficacy of vaccination in cattle populations is highly variable and not well understood [6,7]. Antibody titers are commonly included in veterinary research, but other outcomes, such as respiratory clinical signs, lung lesions, and virus excretion, are also used to evaluate vaccines.

Systematic reviews provide industry professionals and researchers with a critically evaluated summary of available information for decision-making processes as compared to a broader literature review. Research related to vaccination and BRD exists; however, there are very few systematic reviews that provide a consensus on outcomes such as performance and disease. Previous systematic reviews have addressed the effect of respiratory vaccination in the first 45 days in the feedlot on morbidity rates and the impact of respiratory vaccination for bacterial pathogens on morbidity, mortality, and post-mortem lung lesions [6,7]. However, to our knowledge, the evaluation of titer response, a common metric used by researchers and animal health professionals to evaluate disease response and vaccine efficacy, has yet to be evaluated by systematic review [8,9]. The objective of this systematic review was to evaluate the impact of vaccination for respiratory disease on antibody titer responses and health or performance outcomes in beef and dairy cattle.

## 2. Materials and Methods

A comprehensive search of the literature was performed according to PRISMA guidelines using six databases: CAB Abstracts, Ovid MEDLINE, Web of Science Biosis Citation index, Web of Science Core collection, AGRICOLA (EBSCO), and ProQuest Dissertations and Theses [10]. Initial searches were conducted on 10 October 2022. Additionally, hand searches were conducted in The Bovine Practitioner, 1982–10 October 2022, due to certain volumes not being well represented/indexed in the aforementioned search platforms. Hand searching was accomplished by evaluating table of contents for titles relevant to respiratory vaccination and saving them for title abstract screening.

### 2.1. Searches

The search was created by MEC based on previous searches [11,12,13,14,15,16,17,18] employed to ask different but related questions [6]. Original search was reviewed by authors HRM, SFC, and MAS, as content experts, then peer reviewed by LMR using Peer Review of Electronic Search Strategies (PRESS) guidelines [19]. The search was edited according to peer-reviewed input and translated to different databases by MEC. Searches were performed by combining the species, disease, and vaccine concepts for each database. The general search strategy was as follows:Species“Bovine” OR “cattle” OR “bos” OR “bovidae” OR “cow” OR “cows” OR “bovines” OR “heifer” OR “heifers” OR “bull” OR “bulls” OR “steer” OR “steers” OR “calf” OR “calves” OR “herd” OR “farm” OR “ranch”Disease“pneumon*” OR “respiratory disease*” OR “respiratory diseases” OR “shipping fever” OR “undifferentiated fever” OR “Pneumonic Pasteurellosis” OR “BRD” OR “BRDC” OR “bovine respiratory disease complex” OR “bovine respiratory disease” OR “summer pneumon*” OR “enzootic pneumon*” OR “pasturellosis” OR “pasteurellosis” OR “pasteurelloses” OR “pasteurella infection” OR “mannheimiosis” OR “pleuropneumon*” OR “bronchopneumon*” OR “bronchial pneumon*”Vaccine“vaccin*” OR “toxoid” OR “killed” OR “inactivated antigen” OR “modified live” OR “ML” OR “MLV” OR “bacterin” OR “bacterin-toxoid” OR “bacterin toxoid” OR “innoculat*” OR “inoculat*” OR “anti-serum” OR “antiserum” OR “recombinant”Pathogen“Bovine herpesvirus” OR “bovine herpes virus” OR “BHV1” OR “BHV-1” OR “BoHV” OR “BoHV-1” OR “bovine respiratory syncitial virus” OR “bovine syncitial virus” OR “BRSV” OR “RSV” OR “parainfluenza type 3” OR “parainfluenza-3” OR “para influenza type 3” OR “PI-3” OR “PI3” OR “BPIV3” OR “bovine viral diarrheoa virus” OR “bovine viral diarrhea virus” OR “BVDV” OR “BVD” OR “infectious bovine rhinotraceitis” OR “IBRV” OR “IBR” OR “bovine respiratory coronavirus” OR “bovine coronavirus” OR “BcoV” OR “BCV” OR “mannheimia haemolytica” OR “mannheimia heamolytica” OR “mannhemia haemolytica” OR “mannhemia heamolytica” OR “manhemia haemolytica” OR “manhemia heamolytica” OR “pasteurella haemolytica” OR “pasturella haemolytica” OR “pasterella haemolytica” OR “pasterella heamolytica” OR “pasteurella heamolytica” OR “MHA” OR “MH” OR “pasturella multocida” OR “pasteurella multocida” OR “pasterella multocida” OR “P. multocida” OR “PM” OR “histophilus somnus” OR “histophilus somni” OR haemophilus somni” OR “heamophilus somni” OR “haemophilus somnus” OR “heamophilus somnus” OR “pasteurellacea”

All searches were limited to 1982–10 October 2022, due to the majority of present-day commercially available products being approved after 1982, and to manuscripts available in English or French due to the ability of our team to read these two languages only. MEDLINE [20] and CAB Abstracts [21] were searched using the Ovid platform. The search fields MeSH (MEdical Subject Heading), title, and abstract were searched in MEDLINE, and the Subject Headings (CAB Thesaurus), title, and abstract were searched in CAB Abstracts. The Web of Science Core Collection (Search Science Citation Index Expanded; Conference Proceedings Citation Index—Science, Emerging Sources Citation Index; and Book Citation Index) [22] and Biosis Citation Index [23] were searched using the search fields of TS (title, abstract, author keywords, keywords plus). Agricola [24] was searched using the ProQuest platform [25] via search terms title and abstract. All searches can be found in the SearchRXIV database.

### 2.2. Inclusion and Exclusion Criteria

Inclusion criteria were applied at each stage in the screening process (Table 1). Conference proceedings were automatically excluded as they often do not include enough information to evaluate all inclusion criteria. All breeds, sexes, and production stages were included due to commercially available vaccines being labeled for “cattle” and not for a specific breed, sex, or production stage. Challenge studies and natural infection trials that used commercially available products were included if all other criteria were met. Bias was evaluated on an individual study level. Investigators were responsible for evaluating potential sources of bias including randomization, statistical methods, and blinding of subjective outcomes to mitigate the risk of bias. If any of the above statistical considerations were not thoroughly and adequately described, then the study was excluded from the review.

### 2.3. Record Management

After the final searches were completed, files were uploaded into a bibliographic management software (Zotero v. 6.0.30) for duplicate removal. Deduplication was conducted by HRM, comparing information between studies, including but not limited to type of study, DOI, author list, title, ISSN, and other identifiers. After deduplication in the management software, files were uploaded by HRM into an online systematic review tool (Covidence) used to facilitate and streamline the systematic review process. This tool identified additional duplicates that were manually evaluated in a similar fashion.

### 2.4. Study Selection

After deduplication in both programs, records underwent title and abstract evaluation. Two reviewers (HRM and BIR) evaluated each title and abstract for inclusion based on the previously mentioned criteria (Table 1). Both reviewers had to agree on inclusion/exclusion status. If a conflict was present, reviewers came to a verbal consensus before moving to the next stage of screening. If there was not enough information to determine inclusion/exclusion, then the studies were retained for full-text evaluation. Once all conflicts were resolved, two reviewers (HRM and MAS) evaluated each of the remaining studies’ full text for inclusion/exclusion with the same conflict resolution procedures. Once final inclusion decisions were made, data extraction was completed by reviewers (HRM and MAS) with a fillable digital form (Google Form) from each full-text PDF. Data extraction only included original data reported from the authors, not data from the literature that were previously included in the manuscript that may have played a role in the development of the work. The form compiled the following data into a spreadsheet (Microsoft Excel): vaccine (trade name), study population, location, BRD morbidity, performance, pathogens related to antibody titer values, type of control group, beef or dairy animal, production stage, average weight, average age, vaccine type, BRD mortality, and route of vaccine administration.

## 3. Results

Overall, a total of 3020 records were evaluated in title/abstract screening. Subsequently, 466 reports underwent full-text screening, and 101 were included in the review (Figure 1) [9,26,27,28,29,30,31,32,33,34,35,36,37,38,39,40,41,42,43,44,45,46,47,48,49,50,51,52,53,54,55,56,57,58,59,60,61,62,63,64,65,66,67,68,69,70,71,72,73,74,75,76,77,78,79,80,81,82,83,84,85,86,87,88,89,90,91,92,93,94,95,96,97,98,99,100,101,102,103,104,105,106,107,108,109,110,111,112,113,114,115,116,117,118,119,120,121,122,123,124,125]. The authors used the information from the 101 studies to attempt risk of bias analyses. The minimum number of studies to perform these analyses is three; however, due to the large number of commercial vaccine types, production stages, study designs, etc., the authors were unable to conduct any formal risk of bias analyses. All results that follow are descriptive in nature.

### 3.1. Study Characteristics

Of the 101 studies included in this review, 15 studies were conducted in Canada [44,45,48,49,50,51,52,64,87,88,97,112,113,114,117] and 86 were conducted in the United States [9,26,27,28,29,30,31,32,33,34,35,36,37,38,39,40,41,42,43,46,47,53,54,55,56,57,58,59,60,61,62,63,65,66,67,68,69,70,71,72,73,74,75,76,77,78,79,80,81,82,83,84,85,86,89,90,91,92,93,94,95,96,98,99,100,101,102,103,104,105,106,107,108,109,110,111,115,116,118,119,120,121,122,123,124,125]. Appendix A lists the outcomes of interest and general characteristics of each study, including titer values evaluated, performance outcomes, morbidity, and mortality, as well as general information about each study, such as control group used, vaccine type, and where the study was conducted. Challenge and non-challenge studies were separated into categories due to differences between the study designs. A total of 28 challenge studies are included in this review, with 12 of those using beef calves [9,36,42,45,47,55,59,84,101,102,112,114] as the study subjects and 15 studies using dairy cattle [22,23,26,40,41,44,64,68,69,70,98,108,111,113,115]. No statistical comparisons or meta-analyses could be performed across studies due to variation in reported viral challenge strains, different vaccines, and different animal types (Appendix A). Specifically, there were only four instances in which at least three studies utilized the same vaccine[s] (Table 2): Bovi-Shield Gold 5 [41,50,70,81,110,119], Vista 5 SQ [30,32,64,101,108,122], Express 5 [29,56,91,102], and Bovi-Shield Gold One Shot [40,79,80]. However, due to differences in the animal type used and/or type of study conducted, we were unable to perform a meta-analysis for these aforementioned vaccines.

### 3.2. Titer Outcomes

Each study recorded titer outcomes whether the treatment group was related to vaccination, supplementation, or management strategies. Sampling time points included days before vaccination, days after vaccination, prior to and post-challenge, months after vaccination, arrival, first treatment, second treatment, and third treatment plus others (Appendix A). Transformations and different statistical methods in the papers resulted in multiple defined protocols for reporting titer values, where no three studies reported comparable titer variables.

### 3.3. Performance and Health Outcomes

Performance outcomes commonly included were body weight (BW), average daily gain (ADG), feed-to-gain ratio (F:G), and dry matter intake (DMI). However, some studies reported variables such as conception rates, carcass qualities, and behavior qualities (Appendix A). Health outcomes varied greatly as well; Appendix A depicts BRD-specific morbidity and mortality values as reported by each study. Morbidity was reported by 50/101 (49.5%) of studies, and mortality was reported by 43/101 (42.5%). The reported total morbidity ranged from 0% to 100%, with a median of 35.1% (first quartile morbidity of 9% and third quartile morbidity of 64%). Total mortality ranged from 0% to 90%, with a median of 1.4%, first quartile of 0%, and third quartile of 4.4%.

## 4. Discussion

The objective of this review was to understand the relationship between vaccination and performance or health outcomes and reported antibody titers. While there were many studies included in this review, the variability in the outcomes reported and study designs utilized limited our ability to statistically compare reported outcomes. Our original goal was to perform a meta-analysis; however, the heterogeneity across the 101 retained studies reviewed herein made that unattainable.

Previous reviews related to BRD have had a narrower scope of questions asked and inclusion criteria. Capik and others, in 2021, addressed vaccine efficacy related to common bacterial pathogens [6]. Almost 200 full-text articles studies were evaluated, but only five articles were included [6]. The authors cited issues with statistical methods, bias, and continuity of vaccine label data [6]. In the present systematic review, a total of 101 studies were identified, but further comparisons could not be made due to the differences in vaccines, outcome data, and study design. Similar issues have been found in previous research. O’Connor and others, in 2019, conducted a systematic review and network meta-analysis on viral and bacterial vaccines administered at arrival or close to arrival at the feedlot for the control of BRD [7]. Those authors found that the meta-analysis could not support the effectiveness of commercial vaccines in preventing BRD incidence. Additionally, O’Connor and others cited the authors’ previous research—which was confounded due to treatment application—comparing a respiratory vaccine to a control that also received a respiratory vaccine, further concluding that the veterinary community should strive to understand the effect of on-arrival vaccination on health and performance outcomes; in addition, O’Connor and others questioned the practice of on-arrival vaccination due to the conflicting evidence on efficacy and effectiveness [7]. The lack of understanding regarding vaccine efficacy and effect is further demonstrated by this review. Moreover, the remaining knowledge gaps regarding BRD pathogenesis and interactions between viral and bacterial pathogens further complicate our ability to understand the relationship between cattle production outcomes and vaccination against BRD pathogens. While previous reviews addressed similar questions with a narrower scope, the necessity of a review with a broader extent of questions is important to further understand the effect of vaccines on BRD incidence. Another systematic review completed in 2013 by Tripp and others evaluated the use of viral respiratory vaccines in preventative health protocols of cattle [126]. Those authors found there were very few studies which evaluated the effect of a single pathogen on the prevention of BRD [126]. While several commercially available vaccines targeting BRD are multivalent, published understanding is lacking regarding the individual antigens contained in those multivalent vaccines themselves.

While this review includes 101 studies, further analysis of production outcomes was difficult due to the wide variety of reporting methods. Multiple studies in the review used the same vaccines but evaluated non-uniform outcomes. Study population variation also caused issues when aiming for further analysis; comparing across cattle production stages would be inappropriate due to the differences in cattle over time in both immunity and performance outcomes [127]. The immune system of cattle is highly dynamic and develops over time; it would be illogical to compare the vaccine response of a mature cow to that of a pre-weaned beef heifer [128]. Additionally, approximately 75% of all the studies included did not contain a control group (positive or negative). The lack of a control group creates difficulty in interpreting the true effect of vaccination and the desired outcomes such as disease development within the study population. Vaccine trials that contain a negative control, while uncommon, do improve the understanding of the incidence of disease or outcome in the population if an intervention had not been applied.

Previous research has evaluated vaccine efficacy; however, these data are challenging to interpret when making practical decisions regarding vaccine usage within populations of production cattle [129]. Vaccine efficacy research evaluates vaccines in a controlled environment [130]. Colostrum-deprived dairy animals routinely used in vaccine efficacy research very rarely mirror the cattle industry and, therefore, have limited external validity. Cost and logistical constraints are often major factors inhibiting large-scale vaccine efficacy research that mirrors industry-level production systems. Consequently, differences in funding provisions often change the stakeholders directly benefiting from applied vaccine and animal health research, which may influence the outcomes of interest and stated hypothesis of a given study. Within this systematic review, approximately 54% (n = 55) of included studies were funded by industry partners, with the remainder of included studies supported internally through public institutions and resources or unknown sources. While these partnerships are valuable for animal health research, there may be limited incentives to include certain elements, such as negative controls, within a designed study. Additionally, vaccine efficacy studies conducted by industry partners have strict requirements outlined through the guideline VB-GL-3.17 of the Canadian Centre for Veterinary biologics [131] and USDA Veterinary Services Memorandum No. 800.207 [132]. However, academic research is not restricted by these guidelines, which may explain some of the heterogeneity in reported outcomes.

Vaccine effectiveness research evaluates vaccines in “real-world” settings [133]. However, this type of research has many confounding variables which are difficult to control. Due to these constraints, studies applying systems dynamic modeling may help generate more data with applicability in real-world settings. Groves and others, in 2022, discussed how a complex adaptive system influences the ability to control BRD in feeder cattle [134]. The authors state that BRD may be a symptom of the unintended consequences of the cattle production systems that we have yet to determine [134]. Integrative systems-based research is imperative as the segmented nature of cattle production in North American systems may be influencing disease and production loss not observed until the next production stage of an animal’s life. Systems-based approaches are an opportunity to understand and influence the long-term health outcomes of BRD with management decisions to control it. Systems-based approaches to research and uniform reporting of outcomes could be pivotal in understanding how to better control and prevent BRD.

This review highlights the lack of uniform reporting in peer-reviewed cattle vaccine research. Few studies within this review possessed comparable attributes, and several lacked sufficient information to conduct statistical comparisons. New research endeavors, especially those utilizing novel technologies and analytical techniques are crucial; however, the lack of uniform reporting is a major concern. Additionally, the Center of Veterinary Biologics (CVB), Canadian Center for Veterinary biologics (CCVB), and other governing bodies have documents that outline general principles and policies related to potency, efficacy, and safety. These guidelines can help to create more uniform research outcomes that can then be compared and should be followed where possible when designing vaccine studies. Recent research has found that the overall number of published systematic reviews in human medicine is increasing on an annual basis [135]. Moreover, human medicine has numerous standardized operating protocols, but there are fewer widespread, mandated protocols currently used in veterinary medicine. This difference may be one of the major reasons that meta-analyses are difficult to conduct in veterinary medicine. Regardless of the research protocol used, outcomes must be standardized to help further our current understanding before implementation of novel approaches.

Another issue with the peer-reviewed literature that was highlighted by our review was the need for standardized reporting. Standardized outcomes, study methodology, and case definitions would allow for more comparable studies. Options for standardized reporting of outcomes in many kinds of trials exist and are increasingly being adopted. For example, the Strengthening the Reporting of Observational Studies in Epidemiology (STROBE) initiative for studies considered observational in nature, specifically case-control studies, cohort, and cross-sectional studies, aids in creating recommendations for reporting and interpreting published results [136]. The Consolidated Standards on Reporting Trials (CONSORT) for the use in reporting outcomes for randomized controlled trials facilitate the reporting of adequately described methodology and results [137]. In addition, the Animal Research: Reporting of In Vivo Experiments (ARRIVE) guidelines are a checklist of information for inclusion in publications describing animal research [138]. While these are good, standardized reporting guidelines, they do not take into account the issues with standardized methodology and outcomes as observed in this review. The use of standardized reporting methods would increase the ability to compare between studies but ultimately does not fully address discrepancies in outcomes [6,7].

Importantly, we are aware of several limitations arising from this study. The inclusion criteria were limited to the United States and Canada, aiming to encompass similar production types, populations, and management styles. Although there is ongoing respiratory vaccination research related to BRD being conducted worldwide, concerns about heterogeneity guided the decision to narrow the geographical scope. The inclusion of studies from 1980 to 10 October 2022 was chosen due to similar concerns. Notably, conference proceedings and other publications from the non-peer-reviewed literature were excluded. While these sources may contain novel information, they often lack the detailed information necessary for inclusion and may differ from the information presented in later peer-review published works. Moreover, only five databases were utilized. Databases were carefully selected to cover a broad spectrum of relevant journals; however, not all journals are indexed by these databases, and it is possible that other databases may have yielded different results. Additionally, despite tailoring our search terms to capture all the relevant literature to our best ability, the possibility that particular manuscripts could have been inadvertently missed remains.

This review provides evidence for the necessity of uniform methodology and reporting across vaccine trials. With uniform reporting and methodology, systematic reviews, such as this one, could potentially decipher patterns and identify new interpretations that would aid with vaccine innovation and BRD management. The need for prevention methods for BRD will continue to increase as industry and consumer demands for decreased antimicrobial use and precision agriculture tactics increase.

## 5. Conclusions

Systematic reviews are vital for evaluating our understanding of and making evidence-based recommendations for various management practices such as vaccination. However, due to the large amounts of variation between published studies, it was difficult to ascertain the effect of vaccination on antibody titers and health or performance outcomes. Improvement and adoption of standardized outcomes, reporting, and methodology will allow for better, more complete systematic reviews that permit future meta-analyses involving vaccination against BRD pathogens.

## Figures and Tables

**Figure 1 vetsci-11-00599-f001:**
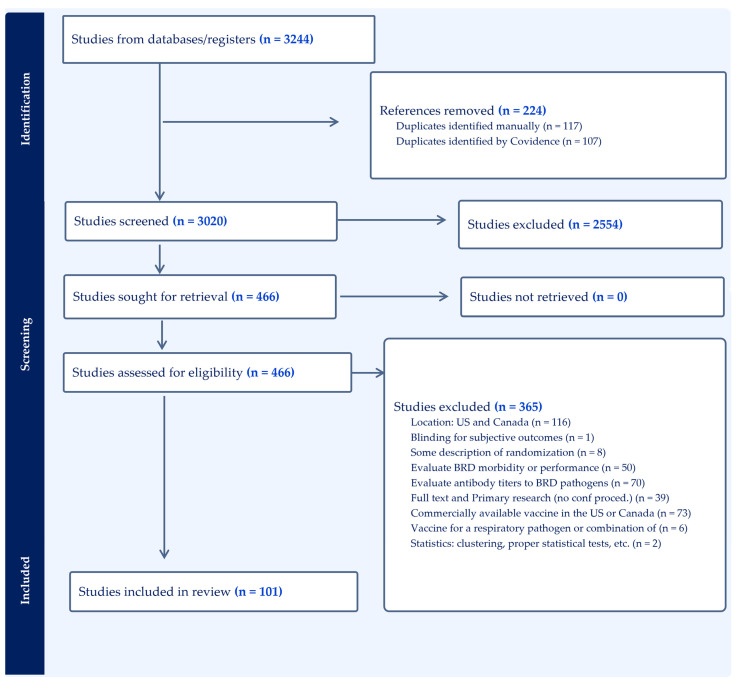
PRISMA flowchart for the systematic review on impact of vaccination for respiratory disease on antibody titer responses, health, and performance in beef and dairy cattle.

**Table 1 vetsci-11-00599-t001:** Inclusion criteria for the systematic review of the impact of respiratory vaccination on antibody titers, health, and performance of beef and dairy cattle.

	Inclusion Criteria
Location	US or Canada
Time	1982–10 October 2022
Population	Beef and dairy cattle
Language	English or French
Resource Type	Full text and primary research (no conference proceedings)
Vaccine type	Commercially available in the US or Canada (extra-label use is acceptable)
Pathogens	Must contain a respiratory pathogen or some combination of Mannheimia haemolytica (Mh), Pasteurella multocida (Pm), Histophilus somni (Hs), bovine viral diarrhea virus (BVDV), bovine herpes virus type1 (BHV1), parainfluenza type 3 (PI3), bovine respiratory syncytial virus (BRSV), or bovine corona virus (BoCoV)
Outcomes Evaluated	Evaluate antibody titers AND BRD morbidity OR performance
Statistical Methods	Blinding for subjective outcomes Some randomization description Sound statistical methods

**Table 2 vetsci-11-00599-t002:** Frequency of commercial vaccine products used in the studies included in the systematic review of the impact of respiratory vaccination on antibody titers, health, and performance of beef and dairy cattle.

Commercial Vaccine Products Used	Frequency in Publications
Bovi-Shield Gold 5	6
Vista 5 SQ	6
Express 5	4
Bovi-Shield Gold One Shot	3
Bovi-Shield Gold One Shot; Bovi-Shield Gold 5; Ultrabac 7	2
Jencine 4	2
PreCon PH	2
Pyramid 5	2
Vista Once SQ	2
20/20 Vision 7 with Spur; Vista 5 SQ	1
Alpha 7; Elite 9-HS	1
Alpha 7; Express 5	1
BarVac 3/Somnugen	1
BoVac	1
Bovine Rhinotracheitis Vaccine, Sanofi Animal Health; Nasalgen IP	1
Bovi-Shield 4; CattleMaster 4; Elite 4; OneShot; Presponce HM; Reliant 3; Triangle 4 PH/HS; Triangle 9 + type II BVD; Triangle 9 PHK; Vira-Shield 5	1
Bovi-Shield 4; Pyramid 4	1
Bovi-Shield BRSV; Bovi-Shield 4; Bovi-Shield 3	1
Bovi-Shield Gold 5; Bovi-Shield Gold One Shot	1
Bovi-Shield Gold 5; Bovi-Shield Gold One Shot; Ultrabac 7	1
Bovi-Shield Gold 5; Bovi-Shield One Shot; Ultrabac 7	1
Bovi-Shield Gold 5; Covexin 8	1
Bovi-Shield Gold 5; One Shot	1
Bovi-Shield Gold 5; One Shot Ultra 7	1
Bovi-Shield Gold One Shot; Ultrabac 7	1
Bovi-Shield Gold One Shot; Ultrabac 7; Bovi-Shield Gold 5	1
Bovi-Shield Gold One Shot; Vira-Shield 6 + Somnus; Clostrishield 7; One Shot Ultra 7; TSV-2	1
Bovi-Shield IBR-PI3; Somnu-Star Ph	1
Bovi-Shield; Bovi Shield 4; Reliant Plus	1
Bovi-Shield; Triangle 3	1
CalfGuard; TSV-4; Bovi-Shield 4; Bovi-Shield Gold 5; Fortress 7; Leptoferm 5; Spirovac	1
Caliber 7; Pyramid 10; Presponse SQ	1
Cattle Master 4; Vira Shield 5; Triangle 4; Premier 4; Tandem 4	1
CattleMaster 4 + L5	1
CattleMaster 4 + L5; Fermicon-7/Somnugen; Pneumo-Guard H	1
CattleMaster 4 + L5; Triangle 9	1
Covexin-8; Express 5	1
Electroid 7; Jencine BVD; Nasalgen (IBR/PI3)	1
Express 5; Inforce 3	1
Express 5; Vision 7	1
Express FP5-VL5	1
IBR-PI3/Somnugen; BRSV Vac; Pioneer Hi-bred Limited; Clostri-Bac 7; Presponse; Coopers IBR-PI3	1
Inforce 3	1
Inforce 3; Bovi-Shield Gold 5	1
Inforce 3; Bovi-Shield Gold FP; Triangle 5	1
Inforce 3; Bovi-Shield Gold FP5; Triangle 5	1
Nasalgen IN; Bar-Vac CSNS	1
Nasalgen IP	1
Nasalgen IP; Bar-Vac CSNS	1
Nasalgen IP; Siteguard M plus Pasteurella	1
Once PMH	1
One Shot Ultra 7; Bovi-Shield Gold 5	1
One Shot Ultra 7; Bovi-Shield Gold 5; Bovi-Shield Gold; Ultrabac 8	1
One Shot; Presponse; Once PMH	1
Onset 5 IN; Once PMH SQ; Vista Once SQ	1
Pneumostar; Somnustar; Somnustar PH	1
Pneumostar; Somnustar; Somnustar PH; Pyramid 4	1
Pyramid 4; Triangle 4	1
Pyramid 5; Triangle 5	1
Pyramid 5; TriVib 5 L; Inforce 3; Ultrabac 8; One Shot BVD	1
Reliant	1
Reliant 4	1
Reliant IBR; Leukotox; Electoid 7	1
Respishield Fusion 4 and IBR 4 way plus	1
Somnu star Ph; BRSV (Bayvet; Etovicoke, Ontario)	1
Somnubac; Ultrabac 7	1
Somubac; Moraxella bovis (Addision Biological Lab.); Bovi-Shield Gold	1
Starvac 4 Plus; Vira Shield 5	1
Synshield; Bovi-Shield 4 + L5	1
Titanium 5; Presponse SQ	1
Triangle 4	1
Triangle 4; Pyramid MLV4; Cattlemaster	1
Triangle 5; Covexin 8; Titanium 5	1
Triangle 9 + Type II BVD; Bovi-Shield 4; One Shot; TSV-2; Pyramid 4; Presponse SQ; CattleMaster 4; Titanium 5; Titanium 5 +PHM Bac 1; Titanium BRSV; Elite 4; Triangle 4; PHK; Triangle 9 +PHK; Titanium IBR; Frontier F3LP Plus; Presponse HM; 4 Once Plus; Express 5 PHM	1
TSV 2	1
TSV-2; Bovi-Shield 5; Vision 7; Endovac-Porci; Pryamid 5	1
TSV2; Nasalgen IP	1
TSV-2; Vision CD-T with SPUR; Bovi Shield 4 + L5; Vision 7; Brucella Abortus Vaccine; Strain RB-51 Live Culture; Presponse HM	1
Ultrabac 8; Pyramid 5; Presponse	1
Vira-Shield 6; Arsenal 4.1; Clostri Shield 7	1
Vista 5 SQ; Vista 3 SQ	1

## Data Availability

The original contributions presented in this study are included in the article/Appendix A. Further inquiries can be directed to the corresponding authors.

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
