# Peer review of "A Systematic Review on the Impact of Vaccination for Respiratory Disease on Antibody Titer Responses, Health, and Performance in Beef and Dairy Cattle"

_vetsci, 2024, doi:10.3390/vetsci11120599_

Round 1

Reviewer 1 Report (Previous Reviewer 2)

Comments and Suggestions for Authors

My previous comments were responded.

Author Response

Reviewer 1: Open Review

Quality of English Language

(x) The quality of English does not limit my understanding of the research.
( ) The English could be improved to more clearly express the research.

Yes

Can be improved

Must be improved

Not applicable

Does the introduction provide sufficient background and include all relevant references?

(x)

( )

( )

( )

Is the research design appropriate?

(x)

( )

( )

( )

Are the methods adequately described?

(x)

( )

( )

( )

Are the results clearly presented?

(x)

( )

( )

( )

Are the conclusions supported by the results?

(x)

( )

( )

( )

Comments and Suggestions for Authors

My previous comments were responded.

Thank you for your previous comments. 

Reviewer 2 Report (New Reviewer)

Comments and Suggestions for Authors

This study attempted to perform a meta-analysis of the effect of vaccination of cattle against bovine respiratory pathogens on antibody titer responses, health, and production outcomes. The authors describe high study heterogeneity and difficulty comparing studies due to the inconsistency of outcome measurements. among trials. At least 2 meta-analyses of the effect of bovine respiratory disease pathogen vaccination on health outcomes have been published in high impact scientific journals in the last 10 years. It is possible that the heterogeneity of studies included in this meta-analysis was a reflection of the variation in the measurement of antibody responses (local, systemic) by different assays in each individual study; however, this reviewer recognizes the need of more consistent reporting guidelines for bovine respiratory disease vaccination efficacy studies.    

Author Response

Reviewer 2

Open Review

Quality of English Language

(x) The quality of English does not limit my understanding of the research.
( ) The English could be improved to more clearly express the research.

Yes

Can be improved

Must be improved

Not applicable

Does the introduction provide sufficient background and include all relevant references?

(x)

( )

( )

( )

Is the research design appropriate?

(x)

( )

( )

( )

Are the methods adequately described?

(x)

( )

( )

( )

Are the results clearly presented?

( )

( )

( )

(x)

Are the conclusions supported by the results?

( )

( )

( )

(x)

Comments and Suggestions for Authors

This study attempted to perform a meta-analysis of the effect of vaccination of cattle against bovine respiratory pathogens on antibody titer responses, health, and production outcomes. The authors describe high study heterogeneity and difficulty comparing studies due to the inconsistency of outcome measurements. among trials. At least 2 meta-analyses of the effect of bovine respiratory disease pathogen vaccination on health outcomes have been published in high impact scientific journals in the last 10 years. It is possible that the heterogeneity of studies included in this meta-analysis was a reflection of the variation in the measurement of antibody responses (local, systemic) by different assays in each individual study; however, this reviewer recognizes the need of more consistent reporting guidelines for bovine respiratory disease vaccination efficacy studies.   

Thank you for your comment. The heterogeneity of the studies which did not allow for us to conduct a metanalysis was based on a variety of factors. The measurement of antibody titers was one example but additionally there were differences in the types of cattle used, vaccines, viral challenge strains, and more which was mentioned on lines 179-183.

Reviewer 3 Report (New Reviewer)

Comments and Suggestions for Authors

Summary: This is an intelligently conceived and written review of vaccines trials for respiratory diseases of cattle (BRD.)  The authors have carefully and critically evaluated published BRD vaccine studies and have highlighted some of the major problems in the related literature.  Inherent problems in these studies include but are not limited to: 1) the lack of repeated studies to demonstrate consistent findings for the same vaccine, 2) dependence on antibody titers and lack of efficacy data based on challenge with specific agents, 3) lack of consistency between experimental designs including subjectivity/objectivity, randomization, statistical analyses, and 4) the wide variety of BRD vaccine products. The authors offer a few recommendations to improve future research on this subject.  Their publication should provide the rationale for more systematic evaluation of the efficacy of BRD vaccines by independent researchers.

Points:

1.       A brief synopsis of the funding sources for these BRD vaccine studies might be revealing. There are rarely instances when these studies are not funded by the biopharma companies which produce the vaccines being tested.

2.       Perhaps the authors have considered the relationship between the administration of vaccines by subcutaneous, intramuscular and even intranasal administration and measurement of systemic (serum) antibodies have with immune defenses in the upper and lower respiratory mucosae. While this question maybe outside of the scope of this paper, it is worth thinking about.

3.       And finally, there is a basic lack of understanding about the pathogenesis of respiratory disease due to the individual microorganisms included in the vaccines and their actual role/causal nature in BRD.  This subject is probably outside the scope of this paper but may again be worth a brief mention.

Recommendations: This manuscript is a welcome addition to the BRD vaccine literature and should be accepted.

Author Response

Reviewer 3

Summary: This is an intelligently conceived and written review of vaccines trials for respiratory diseases of cattle (BRD.)  The authors have carefully and critically evaluated published BRD vaccine studies and have highlighted some of the major problems in the related literature.  Inherent problems in these studies include but are not limited to: 1) the lack of repeated studies to demonstrate consistent findings for the same vaccine, 2) dependence on antibody titers and lack of efficacy data based on challenge with specific agents, 3) lack of consistency between experimental designs including subjectivity/objectivity, randomization, statistical analyses, and 4) the wide variety of BRD vaccine products. The authors offer a few recommendations to improve future research on this subject.  Their publication should provide the rationale for more systematic evaluation of the efficacy of BRD vaccines by independent researchers.

Points:

  1. A brief synopsis of the funding sources for these BRD vaccine studies might be revealing. There are rarely instances when these studies are not funded by the biopharma companies which produce the vaccines being tested.

Thank you. Additional information related to funding sources was added on lines 287-295.

  1. Perhaps the authors have considered the relationship between the administration of vaccines by subcutaneous, intramuscular and even intranasal administration and measurement of systemic (serum) antibodies have with immune defenses in the upper and lower respiratory mucosae. While this question maybe outside of the scope of this paper, it is worth thinking about.

Thank you for your comment. The authors did consider evaluating the impact of the route of administration in this review. However, due to the variety of factors being assessed by this study, there were concerns that including this additional variable might introduce unnecessary complexity.

  1. And finally, there is a basic lack of understanding about the pathogenesis of respiratory disease due to the individual microorganisms included in the vaccines and their actual role/causal nature in BRD. This subject is probably outside the scope of this paper but may again be worth a brief mention.

The authors appreciate your point. We agree that there is a lack of understanding related to the pathogenesis of BRD and the relationship between vaccines. We added further information to this point on lines 255-258.

Recommendations: This manuscript is a welcome addition to the BRD vaccine literature and should be accepted.

Reviewer 4 Report (New Reviewer)

Comments and Suggestions for Authors

The manuscript entitled “A systematic review on the impact of vaccination for respiratory disease on antibody titer responses, health, and performance in beef and dairy cattle” by McAllister et al is a well-written, concise article trying to address the effect of commercially available BRD vaccines and titer response and their correlation with overall health or performance in cattle.

The methodology used (systematic review) is correct and the authors followed the proper current guidelines to achieve this, however regardless of the methodology and results, there are multiple shortcomings in the way this paper is written and formatted.

I have the impression that this article was submitted in a rush, perhaps to fulfill the requirements for graduation of a grad student. Within the text there are multiple duplicated words/syntax incongruencies/underlined text (i.e.  lines 60-62, 182-183, 267, 305-310, 329-333) and this needs to be remediated before publication. Thus, I expect the authors address all these in the next round of review.

In addition, multiple tables are poorly formatted and need modifications. Specifically, in table 1, what does the number “149” represent on the top left square?

Also, table 2 can be modified to better express the data. For example, it indicates is a frequency table so please arrange the table in order of frequency (i.e. vaccines with higher numbers (i.e  n=6) placed first and then follow ordinal arrangement). Individual count vaccines (n=1) should be arranged alphabetically after sorting by frequency. Please review too supplemental tables and follow proper formatting.

In regard to figure 1, please indicate/ connect with another tile, the tile on the lowest right end, as although we later with the text and supplemental materials can interpret this tile, at first glance/read this appears odd and its difficult to interpret.

In line 247 and 248 you provide mortality and morbidity information of studies included in this systemic review. Please would you be able to provide summary descriptive statistics for these indicators (range, five-point statistics, SD, etc)

Finally, please reformat references as multiple styles are observed. In addition, use original references and not those provided by index software (i.e. Medline by ovid, searchRvix).

Author Response

Reviewer 4

The manuscript entitled “A systematic review on the impact of vaccination for respiratory disease on antibody titer responses, health, and performance in beef and dairy cattle” by McAllister et al is a well-written, concise article trying to address the effect of commercially available BRD vaccines and titer response and their correlation with overall health or performance in cattle.

The methodology used (systematic review) is correct and the authors followed the proper current guidelines to achieve this, however regardless of the methodology and results, there are multiple shortcomings in the way this paper is written and formatted.

Thank you for all your comments and each individual comment has a reply.

I have the impression that this article was submitted in a rush, perhaps to fulfill the requirements for graduation of a grad student. Within the text there are multiple duplicated words/syntax incongruencies/underlined text (i.e.  lines 60-62, 182-183, 267, 305-310, 329-333) and this needs to be remediated before publication. Thus, I expect the authors address all these in the next round of review.

Thank you for pointing out these discrepancies. All the previously mentioned issues have been altered.

In addition, multiple tables are poorly formatted and need modifications. Specifically, in table 1, what does the number “149” represent on the top left square?

The number 149 would be the line number for the table however when we open the document that number is not visible. Formatting was corrected and issues appear to have been resolved.

Also, table 2 can be modified to better express the data. For example, it indicates is a frequency table so please arrange the table in order of frequency (i.e. vaccines with higher numbers (i.e  n=6) placed first and then follow ordinal arrangement). Individual count vaccines (n=1) should be arranged alphabetically after sorting by frequency. Please review too supplemental tables and follow proper formatting.

Thank you for your comment. Formatting has been altered. Table 2 is now located on Line 210.

In regard to figure 1, please indicate/ connect with another tile, the tile on the lowest right end, as although we later with the text and supplemental materials can interpret this tile, at first glance/read this appears odd and its difficult to interpret.

Thank you for pointing this mistake out. The correct arrow has been added to improve understanding.

In line 247 and 248 you provide mortality and morbidity information of studies included in this systemic review. Please would you be able to provide summary descriptive statistics for these indicators (range, five-point statistics, SD, etc)

Additional descriptive statistics have been added on lines 227-230.

Finally, please reformat references as multiple styles are observed. In addition, use original references and not those provided by index software (i.e. Medline by ovid, searchRvix).

Thank you for bringing this to our attention. All citations have been reviewed and corrected where needed.  The citations for the searches that this literature search was based off (11-18) and the citations for this search (20-25) are formatted per the databases recommendation. These citations do not have the common components for a citation which led the authors to cite them in the form recommended by the database.

This manuscript is a resubmission of an earlier submission. The following is a list of the peer review reports and author responses from that submission.

Round 1

Reviewer 1 Report

Comments and Suggestions for Authors

The authors have submitted a well-researched review manuscript on the impact of vaccination on disease in cattle.  The review is comprehensive and structured appropriately.  While the review's conclusions were underwhelming, they were not unexpected.  There are no recommended changes to this manuscript.

Reviewer 2 Report

Comments and Suggestions for Authors

In the manuscript of McAllister et al., the authors reported results of a systematic review of the impact of vaccination for respiratory disease on antibody titer responses, health, and performance in beef and dairy cattle. This study was conducted under Prisma 2020 guidelines for systematic reviews and PRESS guidelines utilizing five databases. Criteria for study inclusion were research conducted in the USA or Canada, between 1982 and October 10, 2022, on beef or dairy cattle, using a commercially available vaccine labeled for a respiratory pathogen of interest, and evaluation of antibody titers, and performance or morbidity. A total of 3,020 records underwent title and abstract evaluation. Full‐text analysis was con ducted on 466 reports; 101 studies were included in the final review.  

The aim of this study was to evaluate the impact of vaccination for respiratory disease on antibody titer responses and health or performance outcomes in beef and dairy cattle. This aim was not achieved due to variations between published studies.

The main conclusion of the study is a need for consistent studies reporting. Requirements for reporting of the laboratory and field efficacy vaccine trials are outlined in the guideline VB-GL-3.17 of the Canadian Centre for Veterinary biologics and USDA Veterinary Services Memorandum No. 800.207. The industry is following these guidelines, but academic researchers are usually not.

However, not only uniform reporting is required, but it is important to design and conduct vaccine efficacy studies in accordance with the general principles and requirements of the European Pharmacopoeia (Ph. Eur.) or the Title 9 of Code of Federal Regulations (9CFR) and the guidelines from the Canadian Centre for Veterinary Biologics.

Also, note antibody responses are not primary parameter to evaluate vaccine efficacy against respiratory disease. The main parameters to be assessed to determine the vaccine efficacy are the virus excretion (titre and duration), the general and respiratory clinical signs and lung lesions.